# The first moment of income density functions and estimation of single-parametric Lorenz curves

**Liang Frank Shao***

School of Economics, Henan University, Kaifeng, Henan, China

* mingliangshao@foxmail.com

## Abstract

This paper discusses the first moment, i.e., the mean income point, of income density functions and the estimation of single-parametric Lorenz curves. The mean income point is implied by an income density function and associated with a single-parametric Lorenz function. The boundary of the mean income point can show the flexibility of a parametric Lorenz function. I minimize the sum of squared errors in fitting both grouped income data and the mean income point and identify the best parametric Lorenz function using a large panel dataset. I find that each parametric Lorenz function may do a better job than others in fitting particular grouped data; however, a zero- and unit-modal single-parametric Lorenz function is identified to be the best of eight typical optional functions in fitting most (666 out of 969) observations of a large panel dataset. I perform a Monte Carlo simulation as a robustness check of the empirical estimation.

**Data Availability Statement:** All relevant data are within the manuscript and its Supporting Information files.

**Funding:** The author(s) received no specific funding for this work.

## 1. Introduction

Both the parametric Lorenz curve (LC) function and grouped income data are individually associated with an income distribution probability density function. Krause [1] shows that the modality of the income density function must be considered when fitting grouped income data using the associated Lorenz function. The first moment of an income density function, the mean income point, is often different from the mode. Whether a parametric density function or income data is zero- or unit-modal, the mean income point must be applied as well to evaluate the estimation, which has not been discussed in the related literature. This paper shows that the mean income point of the income probability density functions is another key consideration in estimation. I discuss the first moments of grouped income data and ten typical single-parametric LCs and identify the best parametric LC by minimizing the sum of squared errors (SSE) on both grouped income data and the first moments.

The position of the mean income point of an income density function on the associated LC has been discussed in Shao [2]. It is located at the tangential point of the unit-slope line of the associated LC. In Shao [2], the mean income point is called the mean division point (MDP), and its coordinates are called the mean-income division shares (MDSs), including the mean-

**Competing interests:** The authors have declared that no competing interests exist.

income population share (MPS) and the mean income share (MIS). I use these abbreviations in this study. Fig 1 below shows that point M is the MDP for the LC $\widehat{OMC}$. The diagonal line $\overline{AB}$ has a unit intercept and a slope of negative one, and the dashed line going through point M is parallel to line $\overline{AB}$.

The position of the MDP, point M, can be either on line AB or on one of the two sides of the line when we change the LC by moving point M on the 45˚ line that goes through M. The sum of the MDSs (*MPS + MIS*), denoted by *MDC*, is the intercept of the dashed line going through point M. Because the slope of the dashed line is negative one so that the sum of the coordinates of any point on the line is equal to its intercept. Clearly, we have $MDC \in (0, 2)$. Now, if we change the shape of the LC $\widehat{OMC}$ to let its MDP (point M) be at, below or above line AB, then the *MDC* will be equal to, less than, or larger than one, respectively. Fig 16 in S1 Appendix illustrates the other two cases, MDC ≤ 1, using the Chotikapanich and RGKO LCs (refer to the next subsection).

The boundary of the *MDC* helps to measure the flexibility of a parametric LC. For a perfectly flexible parametric Lorenz function, the MDP should be able to be located at any places in the triangular area ΔOBC in Fig 1. For instance, point M can move along two directions, either the 45˚ tangential line or the dashed perpendicular line going through M. That is, a flexible parametric function can let its MDC take any values in the interval (0, 2) in one direction and can also keep it constant in another direction when we change the shape of the LC by properly changing its parameter values.

Note that the Pietra ratio (Hoover, Robin Hood or Schutz index) can be defined by the difference *MPS − MIS*. Therefore, the correlation between MDC and inequality is bridged by the MDSs, which can be described by the following two equations: MDC + Pietra = 2MPS, MDC − Pietra = 2MIS. That is, MDC and inequality are either positively correlated for a given MIS or negatively correlated for a given MPS.

As noted in Sarabia, Castillo and Slottje (page 752, [3]), the estimator of least squares with parametric functions on grouped data does not guarantee the estimated Gini to satisfy Gastwirth's [4] Gini bounds, because the estimator may not lead to the best parametric function that captures the data's moments; however, once the data's MIS and MPS have been well captured in the estimation then the estimated Gini will meet the bounds. Therefore, including the MPS and MIS in the estimation of least squares is a stricter constraint than Gastwirth's Gini bounds.

I determine the best parametric LC for grouped income data to be the one that performs the best of all optional functions in fitting the data and capturing the first moment (MDP, point M). The least SSE is used to measure the goodness of fit of the estimation.

I choose the best parametric LC by the following three steps. First, I estimate each parametric LC function by minimizing the SSE ($SSE_{LC}$) of grouped income data and select the parametric LC with the smallest $SSE_{LC}$. Second, I calculate the estimated first moments for each LC chosen in the first step and obtain the distance ($SSE_{MDP}$) between the estimated first moments and the data's first moments calculated by the polynomial LC estimator in Shao [2]. Then, I select the parametric LC with the least $SSE_{MDP}$. Last, if the previous two steps lead to the same parametric LC, then that is the best one I choose for the grouped income data; otherwise, the choice changes to focus on either fitting the data or capturing the first moment, or even looking for new optional functions.

I discuss the first moment for ten typical single-parametric LCs in which I propose two new single-parametric functional forms from current multiparametric LC functions, which are associated with zero-modal densities. The seven single-parametric LC functions (Chotikapanich [5]; Gupta [6]; Rohde [7]; the Weibull, the Fisk, log-normal, and Pareto functions) chosen

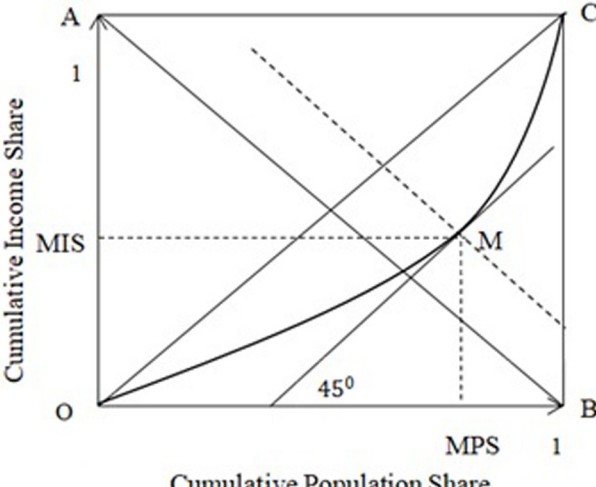

**Fig 1. The mean-income division point M.**

from the literature perform better than the other optional functions (e.g., Krause [1], Ogwang and Tao [8], Hossain and Saeki [9], Wang and Smyth [10], among others) that I have tried and dropped in my estimation. I show that the first moment is a simple and effective factor to evaluate the parametric estimation of grouped income data. Each parametric LC may do a better job than all others in fitting the observations of a particular subset of a large panel of grouped income data; however, a zero-and unit-modal single-parametric LC is identified to perform the best for most (666 out of 969) observations in the panel dataset.

The paper consists of six sections. Section 2 discusses the MDSs, MDC and modality of ten single-parametric LCs, Section 3 describes the data, Section 4 contains the estimation results, Section 5 performs Monte Carlo simulation and Section 6 concludes.

## 2. Single-parametric Lorenz function

After considering as many meaningful single-parametric versions as I can find for all current multiparametric LCs (e.g., Ogwang and Tao [8], Wang and Smyth [10], Hossain and Saeki [9], Abdalla and Hassan [11], Wang, Smyth and NG [12], among others) and the widely cited single-parametric functions, I come up with ten single-parametric LCs which are discussed in this subsection.

Employing the analysis of Shao [2] and theorems 1 and 2 from Krause [1], I present the following two propositions (Proposition 2 is also described in Sarabia, Jorda and Trueba [13]), which are used later in the context to calculate the mean and mode income points on the Lorenz curve.

**Proposition 1** Let $p$ denote the cumulative population share, $y$ be the cumulative income share, $y = L(p)$ be a differentiable LC, $F(x)$ be the cumulative income probability function, and $x$ be an income variable with a mean of $\mu$. I define the cumulative population share at the mean income $\mu$ as $MPS = F(\mu)$. Then, the MDSs can be found by the following equations:

$$(MPS, MIS): \; L'(p)_{p=MPS} = 1, \; MIS = L(MPS).$$

**Proposition 2** For a $C^3$ (third differentiable and continuous) parametric LC, $y = L(p)$, it is associated with a cumulative probability function $F(x)$ and a continuous density function $f(x)$.

The density function has a mode $x_m$ if the following conditions are satisfied:

$$L'''(p_m+) > 0, \quad L'''(p_m-) < 0, \quad x_m = F^{-1}(p_m), \quad y_m = (p_m).$$

Brief proofs for the two propositions are provided in S4 Appendix. For convenience of expression, the following definition is presented for the first moments of the Lorenz curve.

**Definition:** The first moments of (an income density function on its associated) Lorenz curve are the mean division shares (MPS, MIS), MDC and mode $(p_m, y_m)$.

I discuss the first moments (MDSs, MDC, $p_m$, $y_m$) for the following ten single-parametric LCs using the above two propositions.

## 1. The RGKO Lorenz function

I first discuss the following single-parametric version of the biparametric LC by Rasche, Gaffney, Koo, and Obst [14], which is one of the Lamé LCs introduced by Henle, Horton and Jakus [15] and studied by Sarabia, Jordá and Trueba [13]. I denote it as RGKO for convenience of exposition.

$$y = L(p) = [1 - (1 - p)^{\alpha}]^{\frac{1}{\alpha}}, \quad 0 < \alpha \leq 1. \tag{1}$$

Let $L'(p) = 1$ and solve the equation for MPS. Then, the following results are obtained for the MDSs of the RGKO function:

$$MPS = 1 - 2^{-\frac{1}{\alpha}} > 0.5, \quad MIS = 2^{-\frac{1}{\alpha}} < 0.5, \quad MDC = 1. \tag{2}$$

Employing proposition 2, the RGKO LC's third derivative, the mode population share $(p_m)$ and income share $(y_m)$ are as follows:

$$L'''(p) = (1-\alpha)^2 (1-p)^{\alpha-3}[1-(1-p)^{\alpha}]^{\frac{1-3\alpha}{\alpha}}\left[\frac{(\alpha-2)}{(1-\alpha)} + 3(1-p)^{\alpha}\right]$$

$$p_m = 1 - \left[\frac{2-\alpha}{3(1-\alpha)}\right]^{\frac{1}{\alpha}} > 0.333 > y_m = \left(\frac{2}{3}\right)^{\frac{1}{\alpha}}, \quad \alpha < .369.$$

The third derivative of the RGKO function shows that it is associated with a zero- and unit-modal density. The mode equation of the RGKO LC shows that to hold $p_m > y_m$, which is a necessary condition for the mode to exist, we need $\alpha < .369$, which leads to the MDSs restrictions MPS > 0.75 and MIS < .25. Therefore, the RGKO LC could be a good option to fit the grouped income data that imply a zero- or unit-modal density function and satisfy the first moments conditions $0.75 < MPS < 1$, $MIS < 0.25$, and $MDC = 1$, which are generally for those grouped income data with a large income inequality because the Pietra ratio (MPS − MIS) would be larger than 0.5 under the MDSs constraints. Shao (2017) notes that the Gini coefficient is positively correlated with MPS and negatively correlated with MIS in the data.

Fig 2 below shows the MDSs and the mode of the RGKO LC with its parameter $\alpha$. It is easy to see that the RGKO function's first moments have the following restrictions: MPS > 0.5, MIS < 0.5 and $p_m > 0.333 > y_m$. For the Weibull function, its first moments have the restrictions: $p_m < 0.634$, and $MPS \in (0.43, 0.64)$. The mean of the MPS of the data is 0.6367. Therefore, the RGKO function may perform a better estimation than the Weibull function for the observations in which the MPS is larger than 0.64 due to their different restrictions on the first moments. The observations of the MPS are often larger than 0.64 in a large proportion of the dataset.

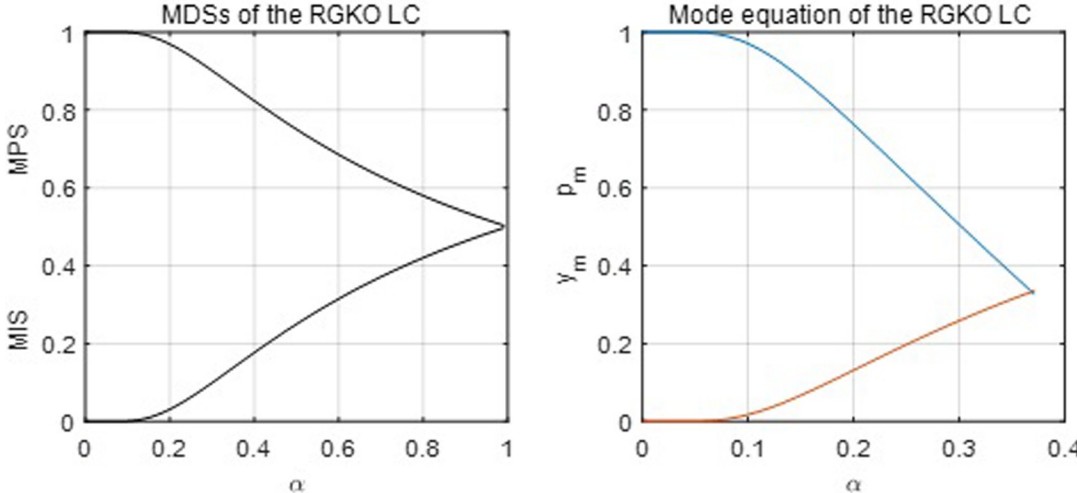

**Fig 2. The MDSs and mode of the RGKO LC.**

## 2. The ASR Lorenz function

Nicholas Rohde [7] proposes a single-parametric LC as follows:

$$y = L(p) = \frac{p(\beta - 1)}{\beta - p}, \ \beta > 1$$

Sarabia, Prieto, and Sarabia [16] find that it is a reparameterization of the function proposed by Aggarwal and Singh [17]. I thus denote the function as ASR. Krause [1] notes that the ASR LC is associated with a zero-modal density function.

The ASR function's MDSs are as follows:

$$\begin{cases} MPS = \beta - \sqrt{\beta(\beta - 1)} > .5, \\ MIS = 1 - \beta + \sqrt{\beta(\beta - 1)} < .5, \end{cases} \quad MDC = 1 \tag{3}$$

The MDCs of the ASR and RGKO functions are one, and the two functions also have the same restrictions for the MDSs. However, as mentioned, their associated density functions have different modalities. The RGKO function can perform much better estimations than the ASR for the data implying unit-modal densities. Fig 3 below shows the MDSs of the ASR LC with its parameter $\beta$.

## 3. The Kakwani Lorenz function

Kakwani [18] proposes a three-parameter LC. I propose a single-parametric version of the function as follows: $y = L(p) = p\left(1 - \delta\sqrt{1 - p}\right), \ \delta \in (0, 1)$.

Let $y'(p) = 1$. Then, the function's MDSs and MDC are as follows:

$$\begin{cases} MPS = \frac{2}{3}, \ MIS = \frac{2}{3}\left(1 - \delta\frac{1}{\sqrt{3}}\right) \in \left(\frac{2(3 - \sqrt{3})}{9}, \frac{2}{3}\right), \\ MDC = \frac{2}{3}\left(2 - \frac{\delta}{\sqrt{3}}\right) \in \left(0.948, \frac{4}{3}\right) \end{cases} \tag{4}$$

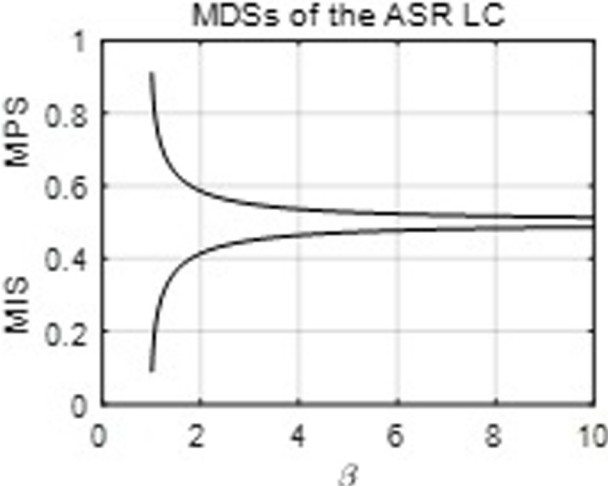

**Fig 3. The MDSs of the ASR LC.**

The third derivative of the function shows that its density function is zero-modal. Fig 4 below shows the MDSs and MDC with parameter $\delta$. Clearly, $MDC \in (.948, 1.333)$ for this LC function.

Thus, the MDC of the Kakwani LC can be either greater or less than one. A limitation of the function is that the MPS is constant, which could be a bad restriction to its capacity to fit the data. However, the mean and standard deviation of the MPS in the disposable income data are 0.64 and 0.046, respectively. Thus, the MPS of the grouped income data is quite stable. It will be shown that the Kakwani LC is still a good option to fit many observations in the data.

## 4. The OMFLG Lorenz function

Ortega, et al. [19] proposed a three-parameter LC. I propose a single-parametric version, denoted as OMFLG, for the function as follows:

$$y = L(p) = p\left[1 - (1 - p)^\theta\right], \quad \theta \in (0, 1].$$

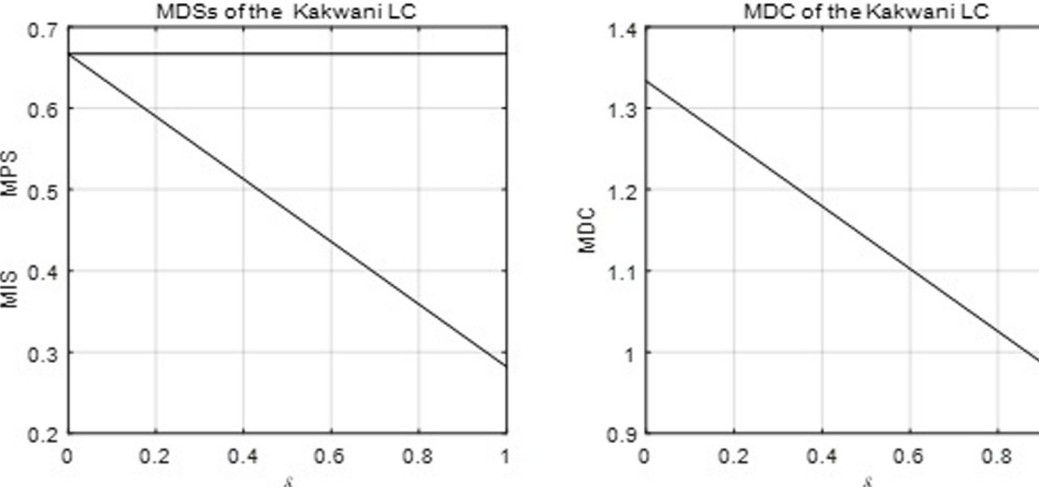

**Fig 4. The MDSs and MDC of the Kakwani LC.**

The function's third derivative is

$$L'''(p) = \theta(1-\theta)(1-p)^{\theta-3}[3-p(\theta+1)], \ \ \theta < 1$$

It is easy to see that the associated density of the OMFLG function is zero-modal since the solution to $y'''(p) = 0$ is $p_m = \frac{3}{\theta+1} > 1, \ \theta < 1$. Let $y'(MPS) = 1$. Then, its MDSs and MDC are solved as follows:

$$\begin{cases} MPS = \dfrac{1}{1+\theta} > .5, \ \ MIS = \dfrac{1}{1+\theta}\left[1 - \left(\dfrac{\theta}{1+\theta}\right)^\theta\right] < .3 \\[4mm] MDC = \dfrac{1}{1+\theta}\left[2 - \left(\dfrac{\theta}{1+\theta}\right)^\theta\right] \in (.75, \ 1.11) \end{cases} \tag{5}$$

Fig 5 below shows the MDSs and MDC of the OMFLG LC with its parameter $\theta$.

## 5. The Chotikapanich Lorenz function

A single-parametric function proposed by Chotikapanich [5] is as follows:

$$y = L(p) = \frac{e^{\sigma p} - 1}{e^\sigma - 1}, \ \sigma > 0.$$

Krause (2014) shows that the Chotikapanich function is associated with a zero-modal density function. The following results are obtained for the MDSs and MDC:

$$\begin{cases} MPS = \dfrac{1}{\sigma}\ln\dfrac{e^\sigma - 1}{\sigma} > .5, \ \ MIS = \dfrac{1}{\sigma} - \dfrac{1}{e^\sigma - 1} < .5, \\[4mm] MDC = \dfrac{1}{\sigma}\ln\dfrac{e^\sigma - 1}{\sigma} + \dfrac{1}{\sigma} - \dfrac{1}{e^\sigma - 1} \in (.8638, \ 1) \end{cases} \tag{6}$$

Fig 6 below shows that its MDC is located in the interval (.8638, 1).

For the hierarchy of exponential Lorenz functions (Sarabia, Castillo and Slottje [3]), $L_1(p; k, \alpha)$, the maximum of its MDC is less than that of the Chotikapanichi function; hence it does

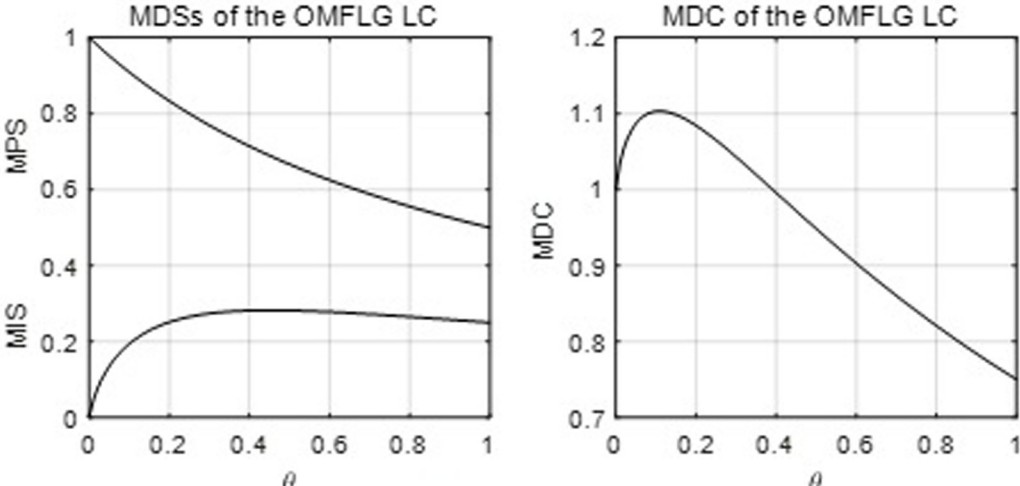

**Fig 5. The MDSs and MDC of the OMFLG LC.**

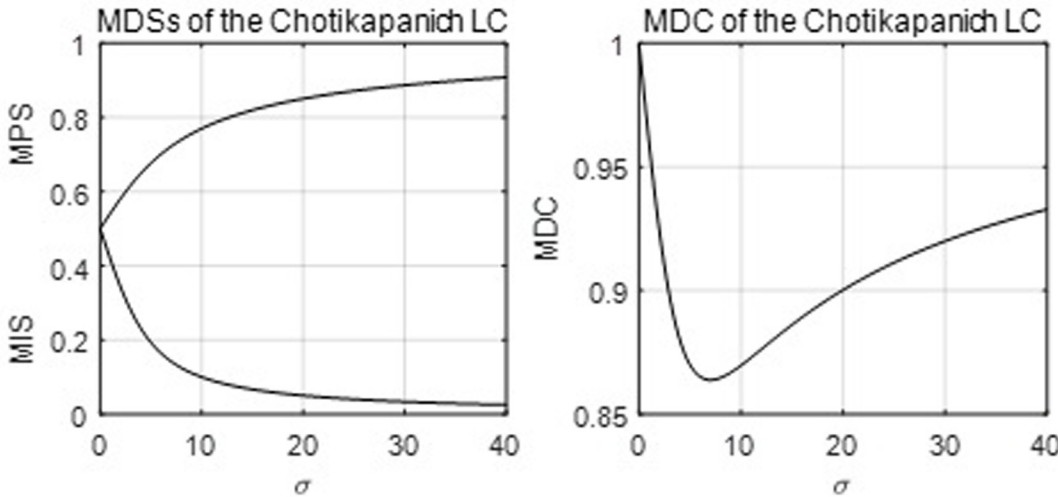

**Fig 6. The MDSs and MDC of the Chotikapanich LC.**

not outperform RGKO, Pareto and Kakwani functions in fitting the Lorenz curves that have an MDC equal or larger than one. The exponential Lorenz function, $L_1(p; k, \alpha)$, does not increase the MPS, MIS and MDC of the initial exponential function ($\frac{e^{\sigma p}-1}{e^{\sigma}-1}$), because both the exponential function and Chotikapanich Lorenz function are convex, and their product shifts toward the left of the two functions. However, in the paper [3] the other two hierarchies of exponential LCs, $L_2(p; k, \gamma)$ and $L_3(p; k, \alpha, \gamma)$, could be more flexible than $L_1(p; k, \alpha)$ in fitting particular data, which are beyond the task of this study.

## 6. The Gupta Lorenz function

Gupta [6] proposes the following single-parametric LC function:

$$y = L(p) = p\tau^{p-1}, \ \tau > 1.$$

The third derivative shows that it is zero-modal. The function does not have closed solutions for the MDSs, and it shows similar properties to the Chotikapanich LC with regard to the MDSs. That is, $MDC < 1, MPS > .5, and MIS < .5$. Table 1 shows the MDSs and MDC for a few values of the parameter $\tau$.

## 7. The Pareto Lorenz function

The Pareto Lorenz curve has the following cumulative distribution function (CDF) with a positive scale parameter $x_m$ and a positive shape parameter $\gamma$:

$$F(x) = \begin{cases} 1 - \dfrac{\gamma x_m^{\gamma}}{x^{\gamma+1}}, & x \geq x_m \\ 0, & x < x_m \end{cases}$$

**Table 1. The first moments of the Gupta LC.**

| $\tau$ | 1.01 | 1.1 | 1.5 | 2 | 2.5 | 5 | 8 | 30 |
|---|---|---|---|---|---|---|---|---|
| MPS | 0.5006 | 0.5058 | 0.5245 | 0.5408 | 0.5528 | 0.5867 | 0.6072 | 0.6554 |
| MIS | 0.4981 | 0.4825 | 0.4325 | 0.3934 | 0.3669 | 0.3018 | 0.2684 | 0.2029 |
| MDC | 0.9987 | 0.9883 | 0.957 | 0.9342 | 0.9187 | 0.8885 | 0.8756 | 0.8584 |

The Pareto Lorenz function is as follows:

$$y = L(p) = 1 - (1-p)^{\frac{1}{\gamma}},\ \gamma > 1.$$

Krause [1] shows that the Pareto function is associated with a zero-modal density function. The following results are obtained for the MDSs and MDC:

$$
\begin{cases}
MDC = 2 - \gamma^{\frac{\gamma}{1-\gamma}} - \gamma^{\frac{1}{1-\gamma}} \in (1,\ 1.2643),\quad \lim_{\gamma \to \infty} MDC = 1 \\[2mm]
MPS = 1 - \gamma^{\frac{\gamma}{1-\gamma}} > .625,\quad MIS = 1 - \gamma^{\frac{1}{1-\gamma}} < .625,
\end{cases}
\tag{7}
$$

Fig 7 below shows the MDSs and MDC of the Pareto LC with parameter $\gamma$.

## 8. The Weibull Lorenz function

Krause [1] discusses the mode of the Weibull distribution, which has the following probability density function with a scale parameter $a$ and a shape parameter $b$:

$$F(x) = 1 - e^{-\left(\frac{x}{a}\right)^b}$$

The Weibull LC function is as follows:

$$y = L(p) = 1 - \frac{\Gamma(-log(1-p),\ 1+b^{-1})}{\Gamma(1+b^{-1})},\ b > 0,\ \Gamma(x,\alpha) = \int_x^\infty t^{\alpha-1}e^{-t}dt$$

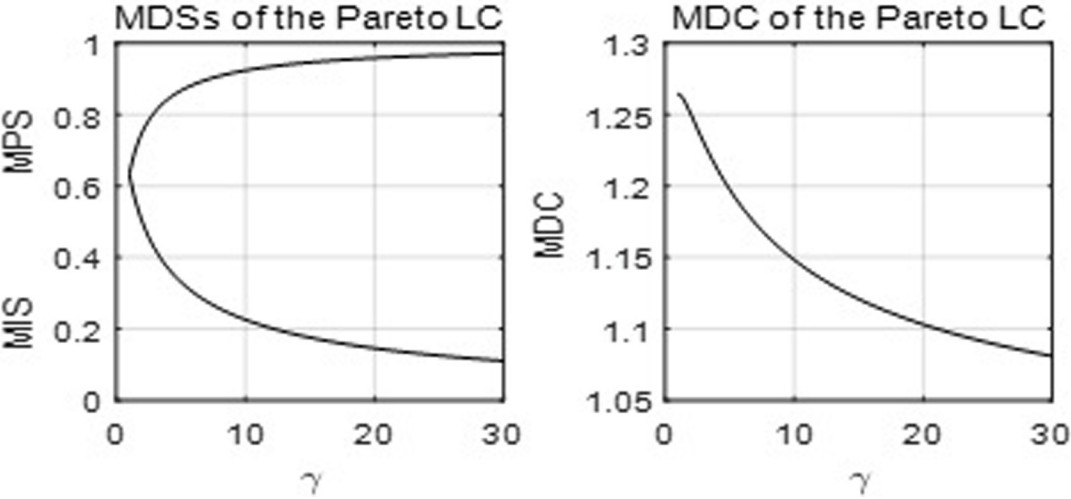

**Fig 7. The MDSs and MDC of the Pareto LC.**

The MDSs and MDC for the Weibull LC are as follows:

$$\begin{cases} MPS = 1 - e^{-\left[\Gamma\left(1+b^{-1}\right)\right]^b} \in (.43, .64), \quad b > 1. \\ MIS = \dfrac{\gamma\left(\left[\Gamma(1+b^{-1})\right]^b, 1 + b^{-1}\right)}{\Gamma(1+b^{-1})} \in (.26, .43) \\ MDC \in (.86, .897). \end{cases} \quad (8)$$

$$L'''(p) = \frac{[-ln(1-p)]^{\frac{1}{b}-2}}{b^2(1-p)^2\Gamma(1+b^{-1})}[1 - b - bln(1-p)],$$

$$p_m = 1 - e^{\frac{1-b}{b}} < .634, \; y_m = 1 - \frac{\Gamma(1-b^{-1}, \; 1+b^{-1})}{\Gamma(1+b^{-1})} < 0.634, \; b > 1.$$

The above equations show that the mode population share $p_m$ of the Weibull density function is less than 0.634, and the corresponding MDC is restricted in the interval (0.86, 0.897), which is not as flexible as the RGKO LC mode and MDC. The MDSs of the Weibull LC take larger values for the MIS and smaller values for the MPS than those of the RGKO LC. Shao [2] notes that a larger Gini coefficient is significantly associated with a larger MPS and a smaller MIS. Therefore, the Weibull function might not be a good option to estimate the data with a large Gini coefficient, but the RGKO function might be the better one to fit those data.

Fig 8 below shows the mode equation, and it is easy to see that $p_m < .634$.

Fig 9 below graphs the MDSs and MDC, and it is easy to see that $MPS > .43 > MIS$ and $MDC \in (.86,1)$.

## 9. The Fisk Lorenz function

The Fisk distribution has the following CDF, which is unit modal as the shape parameter $\beta > 1$.

$$F(x) = \left[1 + (x/\alpha)^{-\beta}\right]^{-1}, \; \alpha > 0, \; \beta > 0$$

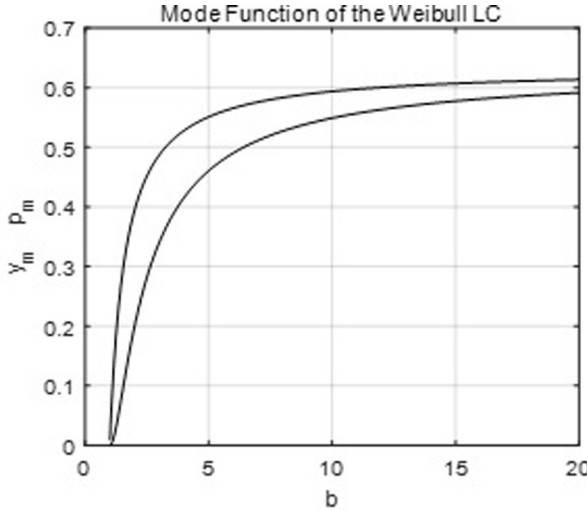

**Fig 8. The parametric mode equation of the Weibull LC.**

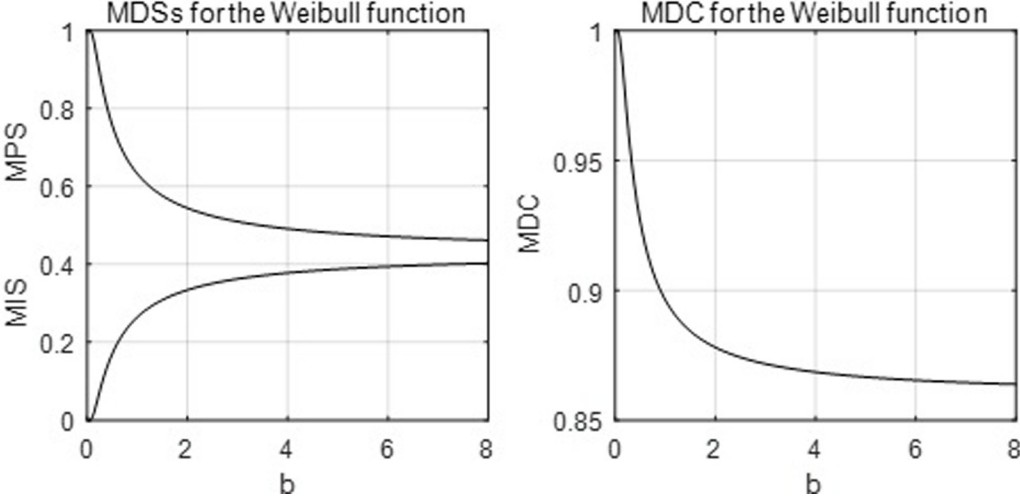

**Fig 9. The MDSs and MDC of the Weibull LC.**

Mean income μ and mode $m$ of the distribution are as follows:

$$\mu = \frac{\frac{\alpha \pi}{\beta}}{sin\left(\frac{\pi}{\beta}\right)}, \quad m = \alpha\left(\frac{\beta - 1}{\beta + 1}\right)^{1/\beta}, \beta > 1$$

Its MPS and mode population share are as follows:

$$MPS = \left[1 + \left(\frac{\pi/\beta}{sin(\pi/\beta)}\right)^{-\beta}\right]^{-1}, \quad p_m = \frac{\beta - 1}{2\beta}, \ \beta > 1$$

The Fisk Lorenz function is as follows:

$$y = L(p) = \frac{sin(\pi/\beta)}{\pi/\beta}\int_0^p \left(\frac{t}{1-t}\right)^{1/\beta} dt$$

Plugging MPS and $p_m$ into the Fisk Lorenz function yields its MIS and $y_m$, respectively. Fig 10 below shows the Fisk MDSs and MDC. Its MDC takes values in the interval (1, 1.09), which is a subset of the interval of the Pareto's MDC. Thus, the Fisk function is not as flexible as the Pareto function regarding the MDC.

Fig 11 below shows the mode population and income shares of the Fisk function. The population share at the mode is less than 0.5, which dramatically restricts the function from capturing the mode of the empirical and simulation datasets because the mode population share is often larger than 0.5 for many observations in the datasets.

## 10. Lognormal Lorenz function

The lognormal distribution has the following CDF:

$$F(x) = \Phi\left(\frac{ln(x) - \mu}{\sigma}\right), \quad \mu > 0, \ \sigma > 0,$$

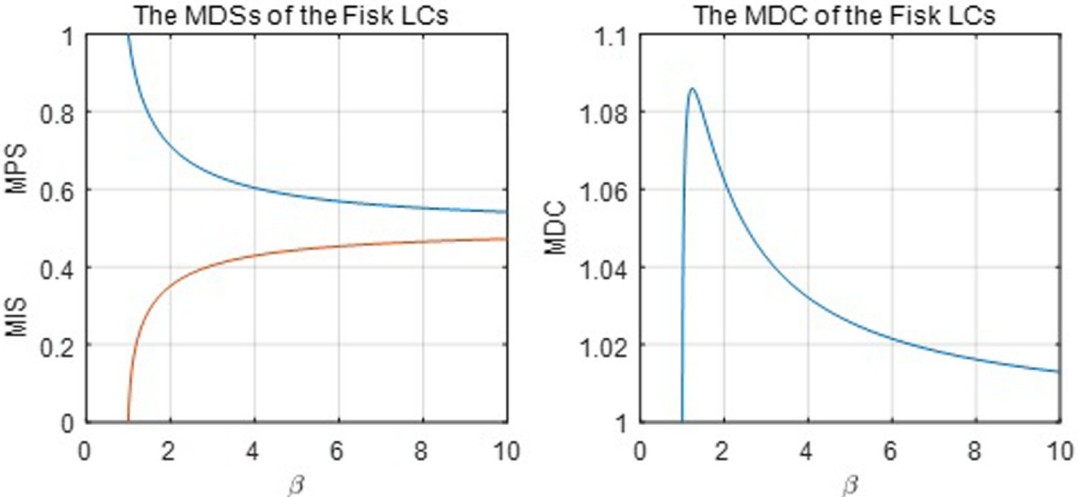

**Fig 10. The MDSs and MDC of the Fisk Lorenz function.**

The distribution is unit modal, and its mean $\mu$ and mode $m$ are as follows:

$$\mu = e^{\mu + 0.5\sigma^2}, \, m = e^{\mu - \sigma^2}$$

Its MDSs and Lorenz function are as follows:

$$y = L(p) = \Phi\big(\Phi^{-1}(p) - \sigma\big),$$

$$MPS = \Phi(0.5\sigma), \quad MIS = \Phi(-0.5\sigma)$$

Its mode population share and income share are as follows:

$$p_m = \Phi(-\sigma), \quad y_m = \Phi(-2\sigma)$$

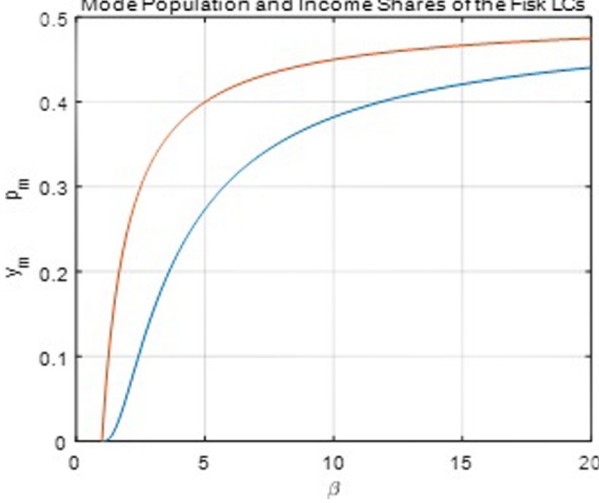

**Fig 11. Mode equations of the Fisk function.**

It is easy to see that its MDC is constant at 1: MDC = 1. Therefore, three functions (RGKO, ASR, Lognormal) have a constant MDC of one, among which empirical and simulation estimations will show that the RGKO function outperforms the other two; meanwhile, the mode population share is less than 0.5, which restricts the function from capturing the mode of the empirical and simulation datasets because the mode population share is often larger than 0.5 for many observations in the datasets. Fig 12 below shows the first moments of the lognormal LC:

Table 2 below summarizes the restrictions of the first moments of the ten single-parametric functions. The table shows that each function has different restrictions on its first moments from what other functions have, which implies the differences in the flexibility of the functions in fitting grouped income data and capturing the first moments.

The Weibull, RGKO, Fisk and Log-normal functions are zero- and unit-modal, and the other six functions are zero-modal. The Weibull function could do a good job in fitting those unit-modal density data that satisfy the conditions MDC $\in$ (0.86, 0.897) and MPS < 0.643, which also restricts its capacity to fit the data because, as the following data subsection shows, the sample mean of the MDC, 1.02, is relatively far away out of the interval, and the mean of the MPS is 0.6367 in the data, which means that approximately half of the observations in the data do not satisfy the MPS restriction of the Weibull function.

There are two related reasons why I focus on these ten functions. One is that I include the latest and widely used single-parametric functions (RGKO, ASR, Pareto, Chotikapanich, Gupta, Weibull, Fisk and lognormal), from which I find that all the MDCs of these functions are either greater or less than one (or equal to one), and none can reach all of the three cases by changing its parameter values. That is, they are not perfectly flexible in the position of MDP; therefore, I am motivated to explore more flexible parametric functions that allow MDC to be both greater and less than one. The other is that I have tried many single-parametric versions of the current multiparametric LCs and ended up with the two functions of OMFLG and Kakwani, which have a more flexible MDP.

Actually, the RGKO LC outperforms all other optional functions in the sense that it can do a better job of both fitting more observations and capturing the first moments, and there are also two reasons for its excellent performance. One is that its MDC is constant at one and that it can be both zero-modal and unit-modal; although the ASR and log-normal functions also

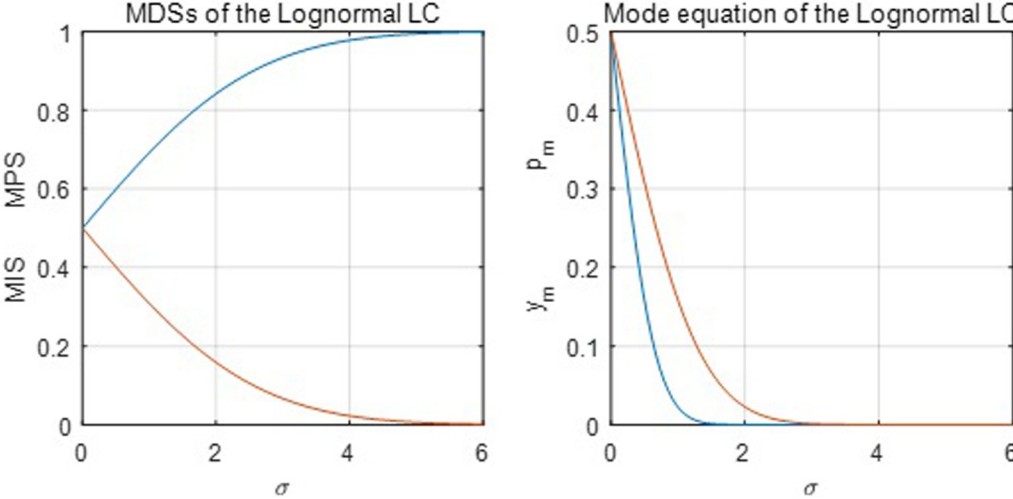

**Fig 12. First moments of the Lognormal LC.**

**Table 2. Summary of the MDSs and modes for the eight single-parametric LCs.**

| Function | Parameter | MPS | MIS | MDC | $p_m$ | $y_m$ |
|---|---|---|---|---|---|---|
| RGKO | $\alpha \in (0.5,1)$ | (0.5, 1) | (0, 0.5) | = 1 | | |
| | (0,0.369) | (0.75, 1) | (0,0.25) | = 1 | (0.333, 1) | (0, 0.333) |
| Weibull | $b > 1$ | (.43, .64) | (.26,.43) | (.86,.897) | (0, 0.634) | (0, 0.634) |
| | $b > 0$ | (0.43, 1) | (0, 0.43) | (0.86, 1) | | |
| ASR | $\beta > 1$ | (0.5, 1) | (0, 0.5) | = 1 | | |
| Kakwani | $\delta \in (0, 1)$ | = 2/3 | (0, 2/3) | (2/3, 4/3) | | |
| OMFLG | $\theta \in (0, 1)$ | (0.5, 1) | (0, 0.3) | (0.75,1.11) | | |
| Chotikapanich | $\sigma > 0$ | (0.5, 1) | (0, 0.5) | (.863, 1) | | |
| Gupta | $\tau > 1$ | (0.5, 1) | (0,0.5) | <1 | | |
| Pareto | $\gamma > 1$ | (0.625, 1) | (0, 0.625) | (1, 1.264) | | |
| Fisk | $\beta > 0$ | (0.5, 1) | (0, 0.5) | (1, 1.09) | (0, 0.5) | (0, 0.5) |
| Lognormal | $\sigma > 0$ | (0.5, 1) | (0, 0.5) | = 1 | (0, 0.5) | (0, 0.5) |

Notes: $\alpha$, $\beta$, $\gamma$, $\delta$, $\theta$, $\sigma$, $\tau$, and $b$ are the parameters to be estimated. The RGKO, Weibull, Fisk and lognormal LCs are zero- and unit-modal, whereas the others are zero-modal.

have the constant unit MDC, the ASR function is not unit modal, and the lognormal function's mode position is more limited than that of the RGKO. Moreover, all other functions can only allow their MDC to be one at one value of their parameters at most (e.g. refer to the MDC curves of the Kakwani and OMFLG functions in Figs 4 and 5). The other is that the data's mean (1.018) of the MDC is very close to one, that is, a large number of observations have unit MDC. Therefore, the RGKO, ASR and lognormal functions match the data best regarding the position of the MDP, but the first moments of the RGKO are the most flexible among the three functions. Similar to the Kakwani LC, its MPS is 2/3, which is very close to the mean (0.636) of the MPS in the dataset, and it is relatively more flexible than the other eight single-parametric functions because its MDC can be both larger and less than one.

In summary, each of the above ten parametric LCs has particular restrictions on its first moments (MDSs, MDC, mode). A prerequisite for a parametric LC to be a good option to fit grouped income data is that their first moments must match each other. That is, the first moments of the density function implied by the data must match the first moments of the parametric LC function. It will be shown in the empirical estimation and simulation subsections that this condition has been almost perfectly satisfied between each function and the subset data of the function.

## 3. Data

I use the data of the MDSs calculated by Shao [2] from a large panel dataset retrieved from WIID4.3 (https://www.wider.unu.edu/project/wiid-world-income-inequality-database). The income variable definition is disposable income per capita. I use the decile data because the goodness of fit may vary for the same observation between the quintile and decile data, and the decile data often provide better accuracy than the quintile data for parametric functions even though there are cases that quintile data occasionally can be better fitted than the decile data by the GMM method (Jordá, Sarabia and Jäntti [20]). The MDS data are estimated using the polynomial estimator in Shao [2]. That is, a polynomial LC function is estimated for each observation using Newton's method; then, the MDSs and MDC are calculated from the estimated polynomial LC.

The polynomial function is supposed to be the most flexible function in function approximation; hence, I assume that the polynomial estimation could capture the empirical first moments of the grouped income data, which might lead to biased estimation and there will be no such concern if the micro data (income data of the density distribution) are available. Fortunately, later simulation shows that the summary statistics of the first moments in both the empirical estimation and simulation are consistent (please refer to Table 8), and the conclusions of the empirical estimation and simulation do not differ. Therefore, the potential bias of the polynomial estimation does not mislead my conclusion.

The decile dataset has 969 observations for 82 countries from 1953 to 2015. The minimum MDC is 0.8443, which is for Norway in 1973, and the maximum MDC is 1.2493, which is for Nepal in 1977. Table 3 summarizes the MDSs and MDC of the dataset. It shows that the first moments seem to be pretty stable because their standard deviations are relatively very small with respect to their mean values, and there are the following inequality relationships, $MIS < MPS < MDC$. Fig 17 in S2 Appendix graphs the observations of the first moments by country.

## 4. Estimation

I apply three steps to identify the best parametric LC in fitting grouped income data and capturing the first moments. First, I use the estimator of nonlinear least squares to estimate the parameter of a single-parametric LC. That is, I solve the following minimization problem for each parametric LC:

$$SSE_{LC} = \min_{x} \sum_{i=0}^{10} \left[ L(p_i, \ x) - y_i \right]^2, \ \ p_0 = y_0 = 0, \ p_{10} = y_{10} = 1.$$

where $L(p_i, x)$ is a single-parametric LC; $x$ is the parameter to be estimated; and $\{(p_i, y_i) : i \in N, 0 \leq i \leq 10\}$ is an observation, which denotes an LC implied by the decile data of an income distribution. I compare the eight values of $SSE_{LC}$ and choose the function with the least $SSE_{LC}$ for each observation. Then, I am able to find one parametric function to best fit every observation. That is, the 969 observations could be divided into eight subsets, each of which can be best fitted by one parametric function in the sense of the least $SSE_{LC}$.

Second, I assume that the estimated MDSs in Shao [2] are the real values of the first moments of the income density implied by grouped income data, and $\widehat{MDSs}$ $\left( \widehat{MPS} \text{ and } \widehat{MIS} \right)$ are the estimated first moments of a parametric LC from the first step. I calculate the standard deviation between $\widehat{MDSs}$ and MDSs and take it as the goodness of fit for a function's performance in capturing the data's first moments.

$$SSE_{MDP} = \sqrt{ \left( MPS - \widehat{MPS} \right)^2 + \left( MIS - \widehat{MIS} \right)^2 }$$

I compare the functions' values of $SSE_{MDP}$ for each observation (LC) and choose the function with the least $SSE_{MDP}$ for the observation. Then, the 969 observations can be partitioned

**Table 3. Summary of the MDSs and MDC data.**

| Variable | Obs. | Mean | Std. Dev. | Min | Max |
| --- | --- | --- | --- | --- | --- |
| MDC | 969 | 1.0186 | 0.0411 | 0.8443 | 1.2493 |
| MPS | 969 | 0.6364 | 0.0458 | 0.5428 | 0.8213 |
| MIS | 969 | 0.3821 | 0.0414 | 0.2067 | 0.4696 |

into eight subsets again, each of which can be best fitted by one function in the sense of the least $SSE_{MDP}$.

Last, I choose the parametric function to be the best option for an observation when the two measures ($SSE_{LC}$, $SSE_{MDP}$) of the goodness of fit lead to the same function. Of course, it is possible that a parametric LC might not be favored using the two measures for any of the observations, and one function might be more favored than the others for some observations.

I proceed with the estimation using the 8 functions in Table 2 excluding the Fisk and log-normal functions because previous discussion shows that the two functions are not as flexible as other functions and inclusion of the two functions does not change our results. Table 4 summarizes the estimation results for the first step of the procedure. The Gupta and Chotikapanich functions are dropped because there are no observations that can be better fitted by the two functions.

The second column of Table 4 is the number of successes, which means the number of observations in which a parametric function always presents a smaller $SSE_{LC}$ than others. The third column shows the restrictions of the MDSs and MDC for each parametric function. The last four columns are the data summary for each subset of the data that are best fitted by the function in each row.

The data statistics in Table 4 show that the first moment restrictions of a parametric LC are almost perfectly satisfied by each subset of the data. The table also shows that the RGKO LC has the best fit in 801 out of 969 observations of the data, and the Kakwani LC follows with 134 observations. There are no observations that can be best fitted by the Gupta and Chotikapanich LCs regarding the goodness of fit $SSE_{LC}$.

The RGKO and ASR LCs have the same restrictions on the first moments, but the RGKO LC performs much better than the ASR LC. The cause of the performance difference might be their modality. That is, if a priori the modality of an income data is unknown a parametric LC

**Table 4. Summary of success according to the minimum $SSE_{LC}$.**

| Function | Success | MDSs | Constraints | Mean | Std. | Max | Min |
|---|---|---|---|---|---|---|---|
| RGKO | | MPS | (0.5, 1) | 0.63 | 0.0436 | 0.7988 | 0.5478 |
| | 801 | MIS | (0, 0.5) | 0.3833 | 0.0407 | 0.4696 | 0.2389 |
| | | MDC | 1 | 1.0133 | 0.0334 | 1.1672 | 0.9195 |
| | | MPS | 0.6667 | 0.6737 | 0.0197 | 0.7243 | 0.6234 |
| Kakwani | 134 | MIS | (0, 0.667) | 0.3902 | 0.0322 | 0.4613 | 0.3168 |
| | | MDC | (0.667, 1.333) | 1.0639 | 0.0286 | 1.1306 | 0.9882 |
| | | MPS | (0.43, 1) | 0.577 | 0.02 | 0.6092 | 0.5428 |
| Weibull | 18 | MIS | (0, 0.43) | 0.3388 | 0.0202 | 0.3764 | 0.3004 |
| | | MDC | (0.86, 1) | 0.9158 | 0.034 | 0.9818 | 0.8443 |
| | | MPS | (0.5, 1) | 0.7149 | 0.0289 | 0.751 | 0.6767 |
| OMFLG | 10 | MIS | (0, 0.3) | 0.2665 | 0.0273 | 0.3051 | 0.2067 |
| | | MDC | (0.75, 1.11) | 0.9815 | 0.0309 | 1.0537 | 0.9469 |
| Pareto | 5 | MPS | (0.625, 1) | 0.7468 | 0.0559 | 0.8213 | 0.649 |
| | | MIS | (0, 0.625) | 0.3638 | 0.0556 | 0.428 | 0.2856 |
| | | MDC | (1, 1.264) | 1.1105 | 0.1004 | 1.2493 | 0.9612 |
| ASR | 1 | MPS | (0.5, 1) | 0.649 | | | |
| | | MIS | (0, 0.5) | 0.3123 | | | |
| | | MDC | 1 | 0.9612 | | | |

Note: The total number of observations is 969. No duplicated observations are counted for different functions' datasets.

with zero- and unit-modal density offers more flexibility and may fit the data much better than those with zero-modal density.

Table 5 summarizes the results for the second step of the procedure. It shows that each parametric function can capture the first moments of some observations better than all others; the RGKO and Kakwani functions still outperform other functions for greater numbers of observations.

Finally, Table 6 below shows the estimation results of the three-step procedure for the eight parametric LCs. The second column, success, represents the number of observations for which a single-parametric LC shows a minimum value in both $SSE_{LC}$ and $SSE_{MDP}$ among the eight functions in fitting the large panel of grouped income data and capturing the first moments. The Gupta and Chotikapanich functions are dropped because they are outperformed by other functions for all observations. It shows that the RGKO and the Kakwani functions are more powerful options than others and that the RGKO LC fits the data the best in the sense of both the grouped income data and the first moments for most (666) of the 969 observations in the data.

The three-step estimation procedure shows that the least SSEs estimated on the income data and first moments may not lead to the same function; however, when a function's first moments are more flexible than others at describing the data's first moments, it could be a better option. The RGKO function is found to fit 801 income observations better than other functions, in which 135 (801 − 666) observations' first moments are not well captured by the function with the least $SSE_{LC}$; it can also capture the first moments better than others for 690

**Table 5. Summary of success according to the minimum $SSE_{MDP}$.**

| Function | Success | MDSs | Constraints | Mean | Std. | Max | Min |
|---|---|---|---|---|---|---|---|
| RGKO | 690 | MPS | (0.5, 1) | 0.6323 | 0.0418 | 0.7523 | 0.57 |
| | | MIS | (0, 0.5) | 0.3854 | 0.0374 | 0.4696 | 0.2556 |
| | | MDC | 1 | 1.0178 | 0.0273 | 1.1053 | 0.9577 |
| Kakwani | 156 | MPS | 0.6667 | 0.6632 | 0.0186 | 0.7148 | 0.6266 |
| | | MIS | (0, 0.667) | 0.3988 | 0.0333 | 0.4613 | 0.312 |
| | | MDC | (0.667, 1.333) | 1.062 | 0.0282 | 1.118 | 0.9566 |
| Chotikapinich | 65 | MPS | (0.5, 1) | 0.5821 | 0.0174 | 0.6135 | 0.5487 |
| | | MIS | (0, 0.5) | 0.3668 | 0.0236 | 0.4131 | 0.323 |
| | | MDC | (0.863, 1) | 0.9488 | 0.0118 | 0.9738 | 0.9195 |
| ASR | 26 | MPS | (0.5, 1) | 0.6936 | 0.053 | 0.751 | 0.5797 |
| | | MIS | (0, 0.5) | 0.3074 | 0.0539 | 0.4434 | 0.2378 |
| | | MDC | 1 | 1.001 | 0.0242 | 1.0591 | 0.9612 |
| OMFLG | 11 | MPS | (0.5, 1) | 0.7 | 0.0203 | 0.7485 | 0.6767 |
| | | MIS | (0, 0.3) | 0.2823 | 0.0105 | 0.3051 | 0.2646 |
| | | MDC | (0.75, 1.11) | 0.9824 | 0.0259 | 1.0537 | 0.9601 |
| Weibull | 9 | MPS | (0.43, 1) | 0.551 | 0.0045 | 0.7402 | 0.5428 |
| | | MIS | (0, 0.43) | 0.3687 | 0.0133 | 0.3354 | 0.2067 |
| | | MDC | (0.86, 1) | 0.9197 | 0.0089 | 0.9492 | **0.8443** |
| Pareto | 7 | MPS | (0.625, 1) | 0.7677 | 0.0334 | 0.8213 | 0.7243 |
| | | MIS | (0, 0.625) | 0.3867 | 0.0292 | 0.4109 | 0.3472 |
| | | MDC | (1, 1.264) | 1.1543 | 0.0461 | **1.2493** | 1.116 |
| Gupta | 5 | MPS | (0.625, 1) | 0.551 | 0.0045 | 0.5578 | 0.5478 |
| | | MIS | (0, 0.625) | 0.3687 | 0.0133 | 0.3804 | 0.3346 |
| | | MDC | <1 | 0.9197 | 0.0089 | 0.9282 | 0.9049 |

Note: The total number of observations is 969. No duplicate observations are counted in different functions' datasets.

**Table 6. Summary of success in fitting the data and capturing the first moments.**

| Function | Success | MDSs/MDC | Constraints | Mean | Std. | Max | Min |
|---|---|---|---|---|---|---|---|
| RGKO | 666 | MPS | (0.5, 1) | **0.6305** | 0.0409 | 0.7523 | 0.57 |
| | | MIS | (0, 0.5) | **0.3862** | 0.0375 | 0.4696 | 0.2556 |
| | | MDC | 1 | **1.0167** | 0.0265 | 1.1053 | 0.9577 |
| Kakwani | 110 | MPS | 0.667 | **0.6703** | 0.0162 | 0.7148 | 0.633 |
| | | MIS | (0, 0.667) | **0.3963** | 0.0307 | 0.4613 | 0.3373 |
| | | MDC | (0.667, 1.333) | **1.0665** | 0.0265 | 1.118 | 1.006 |
| OMFLG | 5 | MPS | (0.5, 1) | **0.6994** | 0.0299 | 0.7485 | 0.6767 |
| | | MIS | (0, 0.3) | **0.283** | 0.0148 | 0.3051 | 0.2646 |
| | | MDC | (0.75, 1.11) | **0.9825** | 0.04 | 1.0537 | 0.9601 |
| Weibull | 6 | MPS | (0.43, 1) | **0.5575** | 0.0177 | 0.5841 | 0.5428 |
| | | MIS | (0, 0.43) | **0.3177** | 0.0122 | 0.3354 | 0.3004 |
| | | MDC | (0.86, 1) | **0.8752** | 0.02 | 0.8968 | 0.8443 |
| Pareto | 4 | MPS | (0.625, 1) | **0.7704** | 0.0357 | 0.8213 | 0.7414 |
| | | MIS | (0, 0.625) | **0.3962** | 0.0287 | 0.428 | 0.3638 |
| | | MDC | (1, 1.264) | **1.1666** | 0.0575 | 1.2493 | 1.1233 |
| ASR | 1 | MPS | (0.5, 1) | **0.649** | | | |
| | | MIS | (0, 0.5) | **0.3123** | | | |
| | | MDC | 1 | **0.9612** | | | |

Note: The total number of observations is 792 and there are 177 observations left because they are not perfectly fitted by any of the eight functions. No duplicated observations are counted for different functions' datasets. The estimated results of the parameters and standardized SSE can be found in Tables 9 and 10 in S5 Appendix, each of which contains only 30 observations to save space; the observations are recorded in Table 11 in S5 Appendix. The full estimation results are available upon request.

observations, in which 34 (690 – 666) income observations are not well fitted with the least $SSE_{MDP}$. However, the RGKO function still performs the best in the estimations because most of the observations (666 out of 969) can be perfectly fitted by this function in the sense of the smallest values for both $SSE_{LC}$ and $SSE_{MDP}$.

Table 6 shows the summary statistics of the six income datasets corresponding to the six functions that each has the smallest $SSE_{LC}$ and $SSE_{MDP}$ on the data. Two points can be drawn from the table. First, a parametric LC function's flexibility in fitting grouped income data could be described by or at least related to the constraints of its first moments. Second, the summary statistics of the data's first moments are located within the constraint intervals of the function's first moments, and they are distinct among the six subsample datasets. Therefore, the first moments of the income data, either estimated from grouped income data using a polynomial estimator or calculated from micro data, can be used to choose the right functions in fitting the data so that both the income data and its first moments are well fitted.

The MPS, MIS and MDC must be considered together when choosing functions because only after having spotted both the MDP and inequality can we know what the LC looks like. The MDC tells the location (>, =, <1) of MDP, and MPS and MIS jointly determine inequality size, which can be measured by their difference (Pietra ratio = MPS – MIS). It is not enough to look at only the match of MDSs between the data and parametric functions because the matched MDSs do not mean that the MDCs could match. Therefore, Table 6 shows that the summary statistics of the three first-moment metrics from each subsample data always best match the constraints of the chosen function.

Figs 13 and 14 below draw graphs between the SSE and the first moments (MDC and MPS), which show that the MDC and MPS are the key factors for the RGKO and Kakwani

functions to be respectively mirrored by the $SSE_{MDP}$ and $SSE_{LC}$. The standardized (square root) SSE is smaller when the MDC data are closer to one for the RGKO LC and so is when the MPS data are closer to 2/3 for the Kakwani LC. The shapes of $SSE_{MDP}$ and $SSE_{LC}$ are similar for the two functions in the figures because the two functions fit both the income data and the first moments well.

The RGKO LC is more powerful than the others in fitting grouped income data and hitting the first moments because its first moments match the data better than the other functions.

In summary, a single-parametric Lorenz function outperforms other options in the least squares estimation if the function's first moments match the data's first moments better than the others do.

## 5. Simulation

I perform a Monte Carlo simulation to generate a set of random LCs for each function and apply the estimation procedure in Section 4 on the generated grouped-income data. The random LCs (grouped income data and the first moments) are calculated from a set of values of the function's parameter that is uniformly distributed in its restricted interval (see Table 2). The estimation is exactly the same as what I do in Section 4; that is, I fit the income and MDSs data using the 8 functions and compare the residuals of one function with the other 7 functions (excluding the Fisk and log-normal functions because inclusion of the two functions does not change our results). Note that the data generated from either LC (grouped income data) or its associated density function (micro data) are equivalent because a Lorenz function and its associated density function express the same income distribution.

Taking the RGKO function as an example, the simulation procedure is as follows. First, we generate a set (e.g., 1000) of random values for the parameter, which is uniformly distributed in (0, 1), and set up the cumulative population shares to be a vector of 11 points including all decile points, 0 and 1. Second, I generate a set of 1000 random RGKO LCs (grouped income data) by the functions (1) and (2). Finally, I perform the same estimation procedure as Section 4 with the simulated data. I repeat the above procedure for the other 7 functions.

For example, for a set of random RGKO LCs including extreme shapes when the parameter α is less than 0.25, some of the income data and their first moments can be better fitted by the

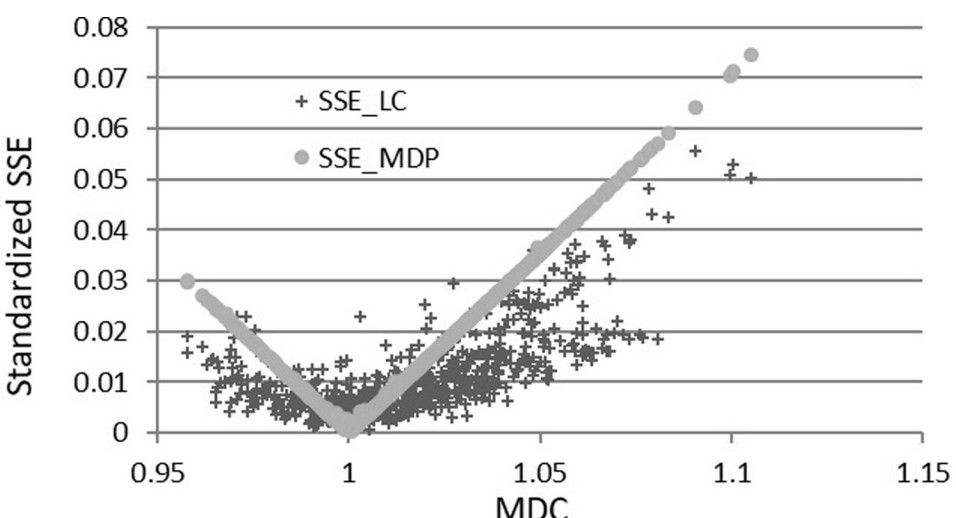

**Fig 13. Standardized $SSE_{MDP}$ and MDC for the RGKO LC (666 observations).**

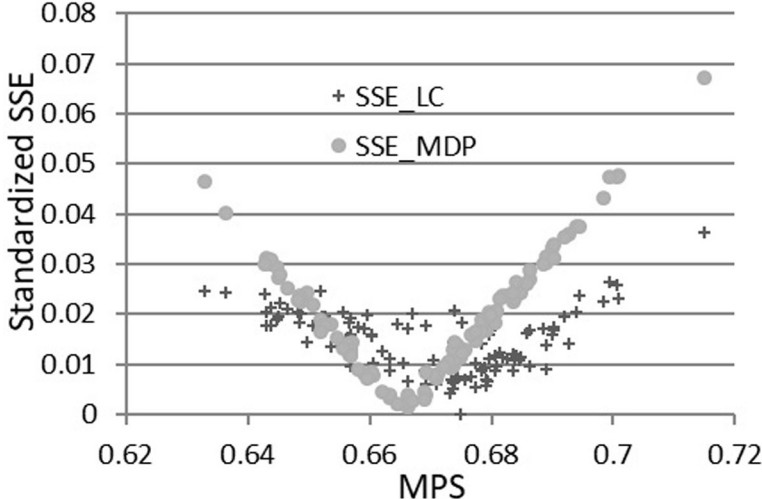

**Fig 14. Standardized $SSE_{MDP}$ and MPS for the Kakwani LC (110 observations).**

other functions (ASR, OMFLG, Pareto and Weibull functions). For the RGKO LCs generated when the parameter α is greater than 0.25, none of the other 7 functions can outperform the RGKO function in fitting both the income data and capturing its first moments; hence, we call the interval (0.25, 1) the critical boundary of the RGKO function in the simulation. Fig 15 shows the two sets of random RGKO LCs: one is for α to be within the critical boundary, and the other is for α to be out of the boundary. Each set has 1000 LCs.

Therefore, there could be extreme cases of a parametric function; the function might not be able to beat out the other functions in fitting these extreme cases of the function. Table 7 summarizes the critical boundaries described by intervals for the 7 functions' parameters. Each function outperforms the other 7 functions in both fitting the data and hitting the first moments that are randomly generated by the parameter to be within the critical boundary. Note that the critical boundaries might be shrunk if more flexible functions are included in the estimation.

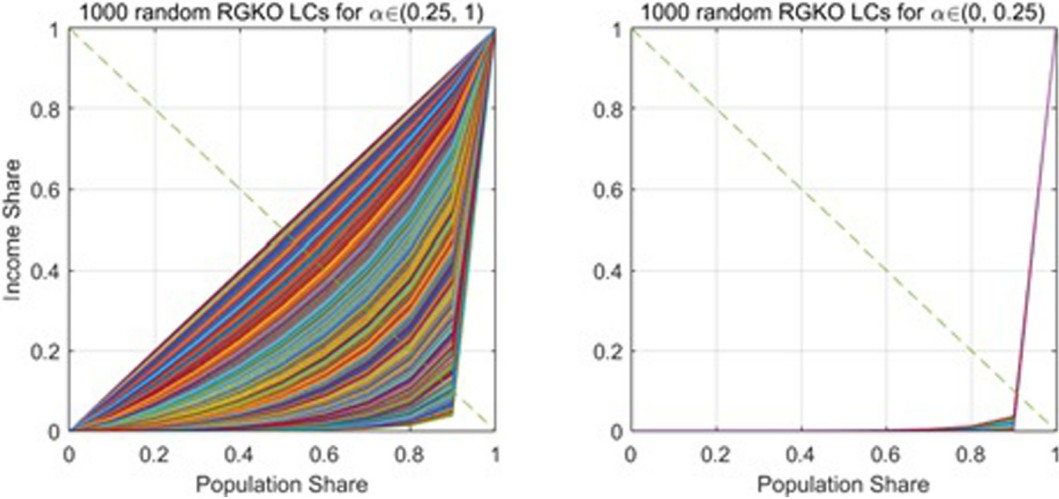

**Fig 15. The RGKO LCs for α to be within and across the critical boundary (0.25, 1).**

**Table 7. Parameter critical boundaries.**

| RGKO | ASR | Kakwani | Pareto | Weibull | OMFLG | Chotikapanich |
|---|---|---|---|---|---|---|
| (0.25,1) | (1,16) | (0.05,1) | (1,198) | (0.2,84.3) | (0,1) | (0,50.3) |

Note: The Monte Carlo simulation for the Gupta function is not performed because it does not have a closed form for its first moments; hence, I do not have its critical boundary interval. Fig 18 in S3 Appendix shows the LCs within the critical boundary and the extreme cases for the other functions. The critical boundary of the lognormal function parameter is (0.06, 3.82) by which it outperforms other functions in fitting the simulated data but the data's first moments can always be better captured by other functions.

Table 8 below summarizes the data statistics for each function in both the simulation estimation and the empirical estimation (see Table 6). Each function outperforms all other functions in both fitting the grouped income data and hitting their first moments in the estimations using the sub data sets associated with the statistics. Table 8 shows that the empirical statistics are always within the limit of simulation statistics (called critical values of the first moments) for each function.

Two points are concluded from Table 8. One is that a parametric function can fit both the data and its first moments very well when the statistics of the data's first moments match the statistics (critical values) of the simulation of the chosen function; therefore, the critical values of the first moments may serve as standard rules to evaluate empirical estimation; the other is that the flexibility of parametric LC functions in fitting a dataset could be explained by the size of standard deviation of the first moments in the Monte Carlo simulation of the functions. For

**Table 8. Summary statistics of the simulation and empirical estimation.**

| | | Simulation Statistics | | | | Empirical Statistics | | | |
|---|---|---|---|---|---|---|---|---|---|
| | | MEAN | MAX | MIN | STD. | MEAN | MAX | MIN | STD. |
| RGKO | MPS | 0.691 | 0.937 | 0.500 | **0.130** | 0.631 | 0.752 | 0.570 | **0.041** |
| | MIS | 0.309 | 0.500 | 0.063 | **0.130** | 0.386 | 0.470 | 0.256 | **0.038** |
| | MDC | 1.000 | 1.000 | 1.000 | 0.000 | 1.017 | 1.105 | 0.958 | 0.027 |
| OMFLG | MPS | 0.691 | 0.997 | 0.500 | **0.137** | 0.699 | 0.749 | 0.677 | 0.030 |
| | MIS | 0.253 | 0.282 | 0.018 | 0.045 | 0.283 | 0.305 | 0.265 | 0.015 |
| | MDC | 0.944 | 1.103 | 0.750 | **0.114** | 0.983 | 1.054 | 0.960 | 0.040 |
| Pareto | MPS | 0.981 | 0.995 | 0.662 | 0.033 | 0.770 | 0.821 | 0.741 | 0.036 |
| | MIS | 0.069 | 0.601 | 0.026 | 0.072 | 0.396 | 0.428 | 0.364 | 0.029 |
| | MDC | 1.050 | 1.263 | 1.022 | 0.040 | 1.167 | 1.249 | 1.123 | 0.058 |
| Weibull | MPS | 0.448 | 0.913 | 0.433 | 0.045 | 0.558 | 0.584 | 0.543 | 0.018 |
| | MIS | 0.414 | 0.427 | 0.059 | 0.036 | 0.318 | 0.335 | 0.300 | 0.012 |
| | MDC | 0.862 | 0.971 | 0.860 | 0.009 | 0.875 | 0.897 | 0.844 | 0.020 |
| Kakwani | MPS | 0.667 | 0.667 | 0.667 | 0.000 | 0.670 | 0.715 | 0.633 | 0.016 |
| | MIS | 0.458 | 0.647 | 0.282 | **0.104** | 0.396 | 0.461 | 0.337 | 0.031 |
| | MDC | 1.125 | 1.314 | 0.949 | **0.104** | 1.067 | 1.118 | 1.006 | 0.027 |
| ASR | MPS | 0.530 | 0.946 | 0.508 | 0.040 | 0.649 | | | |
| | MIS | 0.470 | 0.492 | 0.054 | 0.040 | 0.312 | | | |
| | MDC | 1.000 | 1.000 | 1.000 | 0.000 | 0.961 | | | |
| Chotikapanich | MPS | 0.835 | 0.922 | 0.500 | 0.096 | | | | |
| | MIS | 0.075 | 0.499 | 0.020 | 0.089 | | | | |
| | MDC | 0.911 | 1.000 | 0.864 | 0.026 | | | | |

Note: The data size for each function's simulation is 1000 observations. Highlighted values are the relatively large ones of the standard deviation among the 7 functions.

instance, RGKO has the best flexibility in its simulated MDSs because the standard deviation (0.13 for both MPS and MIS) of its simulated MDSs is the largest among the 7 functions.

In summary, the flexibility of parametric LC functions in fitting a dataset could be explained by the size of the standard deviation of their first moments in the simulation.

## 6. Concluding remarks

Grouped income data are associated with an income probability density function. A parametric Lorenz function also implies a probability density function. Therefore, the first moments of the two density functions must be well matched when using a parametric Lorenz function to fit the data; that is, the first moments of parametric Lorenz functions must be considered to evaluate the estimation. The least SSE of the estimated first moments can be used to measure the goodness of fit when comparing different parametric functions.

The single-parametric RGKO function outperforms the other optional single-parametric functions in the empirical estimation, which is confirmed by it having the largest standard deviation of its MDSs in Monte Carlo simulation. The flexibility of a parametric LC can be described by the standard deviation of its first moments in the simulation, which is associated with a critical boundary of its parameter, and the critical boundary also determines the function's capacity in both fitting the data and capturing the data's first moments.

Nonetheless, it is possible to find new single-parametric LCs to outperform the RGKO LC in fitting the data with unit MDC because the first moments are necessary but not sufficient factors to describe the income density function and grouped income data perfectly, and higher-degree moments could also be considered in the estimation.

It is complex to choose the best one from multiple different single-parametric functions in fitting grouped income data. Multiparametric LC functions are often more flexible than single-parametric functions, and it is likely to simplify the work to find one at the cost of more complicated functional forms, which warrant future work.

## Supporting information

**S1 Appendix.**
(DOCX)

**S2 Appendix.**
(DOCX)

**S3 Appendix.**
(DOCX)

**S4 Appendix.**
(DOCX)

**S5 Appendix.**
(DOCX)

**S1 File.**
(RAR)

## Acknowledgments

The author would like to thank two anonymous referees, Markus Jäntti, Melanie Krause, Tsunfeng Chiang, and participants at the Econometrica Society 2018 conferences at Xiamen University, the China Economics Society 2019 conference at Dongbei University of Economics

and Finance, and the CEC 2018 Summer Conference at the Jiangxi University of Economics and Finance for their informative comments.

## Author Contributions

**Conceptualization:** Liang Frank Shao.

**Data curation:** Liang Frank Shao.

**Formal analysis:** Liang Frank Shao.

**Funding acquisition:** Liang Frank Shao.

**Investigation:** Liang Frank Shao.

**Methodology:** Liang Frank Shao.

**Project administration:** Liang Frank Shao.

**Resources:** Liang Frank Shao.

**Software:** Liang Frank Shao.

**Supervision:** Liang Frank Shao.

**Validation:** Liang Frank Shao.

**Visualization:** Liang Frank Shao.

**Writing – original draft:** Liang Frank Shao.

**Writing – review & editing:** Liang Frank Shao.

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
