## [Decision Letter · Decision Letter 0]

24 Jan 2022

PONE-D-21-23351The First Moment of Income Density Functions and Estimation of Single-Parametric Lorenz CurvesPLOS ONE

Dear Dr. Shao,

Thank you for submitting your manuscript to PLOS ONE. After careful consideration, we feel that it has merit but does not fully meet PLOS ONE’s publication criteria as it currently stands. Therefore, we invite you to submit a revised version of the manuscript that addresses the points raised during the review process.

Though a reviewer advise us to reject the manuscript, the reviewer provided many valuable and constructive comments. Considering three reviewers' useful comments and the interesting topic of the manuscript, I would like to give you a chance to revise your manuscript during the special period. The revised manuscript will undergo the next round of review by the same reviewers.

We look forward to receiving your revised manuscript.

Kind regards,

Baogui Xin, Ph.D.

Academic Editor

PLOS ONE

Journal Requirements:

3. Please ensure that you refer to Figures 16 and 17 in your text as, if accepted, production will need this reference to link the reader to the figure.

4. We note you have included a table to which you do not refer in the text of your manuscript. Please ensure that you refer to Table 11 in your text; if accepted, production will need this reference to link the reader to the Table.

Reviewers' comments:

Reviewer's Responses to Questions

**Comments to the Author**

1. Is the manuscript technically sound, and do the data support the conclusions?

Reviewer #1: Yes

Reviewer #2: Yes

Reviewer #3: Partly

2. Has the statistical analysis been performed appropriately and rigorously? 

Reviewer #1: Yes

Reviewer #2: Yes

Reviewer #3: I Don't Know

3. Have the authors made all data underlying the findings in their manuscript fully available?

Reviewer #1: Yes

Reviewer #2: Yes

Reviewer #3: Yes

4. Is the manuscript presented in an intelligible fashion and written in standard English?

Reviewer #1: Yes

Reviewer #2: Yes

Reviewer #3: Yes

5. Review Comments to the Author

Reviewer #1: The scope and content of the paper is technically good. The authors have done a good job in providing a comprehensive analysis on the the mean income point, of income density function and estimation of single-parametric Lorenz curves.

Reviewer #2: Review Report: PONE-D-21-23351

Topic: The First Moment of Income Density Functions and Estimation of Single-Parametric Lorenz Curves

The first moment, the mean income point, of the income density function, and the single-parametric

Lorenz curve estimates are discussed in this study. The flexibility of a parametric Lorenz function can be

seen in the border of the mean income point. Using a large panel data set, the authors found the optimum

parametric Lorenz function by minimizing the sum of squared errors in fitting both grouped income data

and the mean income point. A zero- and unit-modal single-parametric Lorenz function is identified

to be the best of eight typical optional functions in fitting most observations of a large panel data

set. A Monte Carlo simulation is also performed as a robustness check of the empirical

estimation. Following are my specific comments regarding the manuscript:

• Since there is only one author, it will be more appropriate to imply the First person

narrative (singular) throughout the manuscript.

• The basic functions such as Pietra ratio, MPS, MDCs etc should be defined at the initial

stage of the article.

• Like Fisk and lognormal Lorenz function (9 & 10), Weibull and Pareto (7 & 8) must also

be defined clearly in order to maintain uniformity in defining the basic terminologies and

functional forms.

• Why didn’t the author consider the Hierarchical Families of Lorenz Curves* which

perform much better than Kakwani Lorenz function and gives a robust performance in

fitting actual income data across countries?

[Reference: An Exponential Family of Lorenz Curves Author(s): José-María Sarabia,

Enrique Castillo and Daniel J. Slottje Source: Southern Economic Journal,

Vol. 67, No. 3 (Jan., 2001), pp. 748-756 ]

• Gastwirth's (1972) Gini bounds** are non-parametric constraints that should be satisfied

by the Gini index of any parametric family of Lorenz curves. Can it be concluded from

the present study that Gini indices satisfy Gastwirth's bounds, since Gini coefficient is

positively correlated with the MPS and negatively correlated with the MIS in the data?

[Reference: Gastwirth,Joseph L . 1972. The estimation of the Lorenz curve and Gini

index. Review of Economics and Statistics 54 : 306-16.]

• It is strongly suggested to summarize the main findings in section 4 & 5 in the form of

single structured sentence instead in a paragraph. The current representation of findings is

all very confusing.

• It would be interesting to report the current results in comparison to bi-parametric

families.

• The author concluded, “The single-parametric RGKO function outperforms the other

optional single-parametric functions in the empirical estimation of group data.” Can

similar conclusion be drawn to Micro data?

Reviewer #3: ∙ There is little or no theoretical development. Table 2, p.18, is potentially a useful summary of associated Lorenz curve statistics. There is an error in the Proof for Proposition 2, p.38: f′(x)=-L′′′(F(x))f(x)/μL′′(F(x))².

∙ The paper would benefit from some theoretical justification of the selection criteria employed, e.g., the equal weighting employed in SSE_{LC}. Similarly should weighted nonlinear least squares be used in parameter estimation?

∙ The paper concentrates on one parameter Lorenz curve specifications. Is this justified?

6. PLOS authors have the option to publish the peer review history of their article (what does this mean?). If published, this will include your full peer review and any attached files.

Reviewer #1: No

Reviewer #2: No

Reviewer #3: No

---

## [Author Response · Author response to Decision Letter 0]

8 Mar 2022

Dear Reviewers and editor, 

I appreciate your comments and offering me the chance to revise my manuscript. I have uploaded the files including my revised version and answer to the comments. Please let me know if furhter revisions are needed. 

Best regards,

Frank

---

## [Decision Letter · Decision Letter 1]

18 Apr 2022

The First Moment of Income Density Functions and Estimation of Single-Parametric Lorenz Curves

PONE-D-21-23351R1

Dear Dr. Shao,

We’re pleased to inform you that your manuscript has been judged scientifically suitable for publication and will be formally accepted for publication once it meets all outstanding technical requirements.

Kind regards,

Baogui Xin, Ph.D.

Academic Editor

PLOS ONE

Additional Editor Comments (optional):

Reviewers' comments:

Reviewer's Responses to Questions

**Comments to the Author**

1. If the authors have adequately addressed your comments raised in a previous round of review and you feel that this manuscript is now acceptable for publication, you may indicate that here to bypass the “Comments to the Author” section, enter your conflict of interest statement in the “Confidential to Editor” section, and submit your "Accept" recommendation.

Reviewer #1: All comments have been addressed

Reviewer #2: All comments have been addressed

2. Is the manuscript technically sound, and do the data support the conclusions?

Reviewer #1: Yes

Reviewer #2: Yes

3. Has the statistical analysis been performed appropriately and rigorously? 

Reviewer #1: Yes

Reviewer #2: Yes

4. Have the authors made all data underlying the findings in their manuscript fully available?

Reviewer #1: Yes

Reviewer #2: Yes

5. Is the manuscript presented in an intelligible fashion and written in standard English?

Reviewer #1: Yes

Reviewer #2: Yes

6. Review Comments to the Author

Reviewer #1: (No Response)

Reviewer #2: The paper title "The First Moment of Income Density Functions and Estimation of

Single-Parametric Lorenz Curves" with Abstract: This paper discusses the first moment, i.e., the mean income point, of income

density functions and the estimation of single-parametric Lorenz curves. The mean income

point is implied by an income density function and associated with a single-parametric Lorenz

function. The boundary of the mean income point can show the flexibility of a parametric

Lorenz function. I minimize the sum of squared errors in fitting both grouped income data

and the mean income point and identify the best parametric Lorenz function using a large

panel dataset. I find that each parametric Lorenz function may do a better job than others in

fitting particular grouped data; however, a zero- and unit-modal single-parametric Lorenz

function is identified to be the best of eight typical optional functions in fitting most (666 out

of 969) observations of a large panel dataset. I perform a Monte Carlo simulation as a

robustness check of the empirical estimation.

is now acceptable.

7. PLOS authors have the option to publish the peer review history of their article (what does this mean?). If published, this will include your full peer review and any attached files.

Reviewer #1: No

Reviewer #2: No

---

## [Editor Report · Acceptance letter]

14 Jun 2022

PONE-D-21-23351R1 

The First Moment of Income Density Functions and Estimation of Single-Parametric Lorenz Curves 

Dear Dr. Shao:

I'm pleased to inform you that your manuscript has been deemed suitable for publication in PLOS ONE. Congratulations! Your manuscript is now with our production department. 

Kind regards, 

on behalf of

Professor Baogui Xin 

Academic Editor

PLOS ONE